# Efficient Post-Processing for Equal Opportunity in Fair Multi-Class Classification

## Abstract

Fairness in machine learning is of growing concern as more instances of biased model behavior are documented while their adoption continues to rise. The majority of studies have focused on binary classification settings, despite the fact that many real-world problems are inherently multi-class. This paper considers fairness in multi-class classification under the notion of parity of true positive rates—an extension of binary class equalized odds [23]—which ensures equal opportunity to qualified individuals regardless of their demographics. We focus on algorithm design and provide a post-processing method that derives fair classifiers from pre-trained score functions. The method is developed by analyzing the representation of the optimal fair classifier, and is efficient in both sample and time complexity, as it is implemented by linear programs on finite samples. We demonstrate its effectiveness at reducing disparity on benchmark datasets, particularly under large numbers of classes, where existing methods fall short.

## 1 Introduction

Algorithmic fairness has emerged as a topic of significant concern in the field of machine learning, due to the potential for models to exhibit discriminatory behavior towards historically disadvantaged demographics [9, 4, 6], all while their adoption continues to rise in domains including high-stakes areas such as criminal justice, healthcare, and finance [3, 7]. To address the concern, a variety of fairness criteria have been proposed (e.g., demographic parity, equalized odds) along with mitigation methods [10, 19, 23, 26]. On classification problems, the majority of work focuses on the binary class setting [2, Table 1], where one class is typically considered to be more favorable (e.g., the approval vs. rejection of a credit card application).

Yet, many real-world problems are multi-class in nature. In the case of credit card applications, issuers may opt to assigning higher-tier interest rates to high-risk applicants rather than outright rejecting them, which creates opportunities to applicants who would otherwise be denied credit and also generates returns for the banks. Similarly, in online advertising, recruiting platforms can employ machine learning models to match users to relevant job postings across multiple occupation categories. There are evidences, however, for such systems to exhibit gender bias [8, 13, 44]; for instance, models that are trained to identify occupation from biography tend to show higher accuracy (recall) on male biographies than on their female counterparts in occupations that are historically male-dominated [14].

In the example above, unfairness is manifested in a disparity of *true positive rates* (TPRs) across demographic groups $A$ (generalizing the true positive and negative rates in binary classification),

$$\mathrm{TPR}_a(\widehat{Y})_y := \mathbb{P}(\widehat{Y} = y \mid Y = y, A = a), \quad \forall y \in [k], a \in [m].$$

A classifier satisfying parity of TPRs, i.e., $\mathrm{TPR}_a = \mathrm{TPR}_{a'}$ for all $a, a'$, ensures that individuals with the same qualification ($Y$) will have *equal opportunity* of receiving their favorable outcome ($\widehat{Y} = Y$)

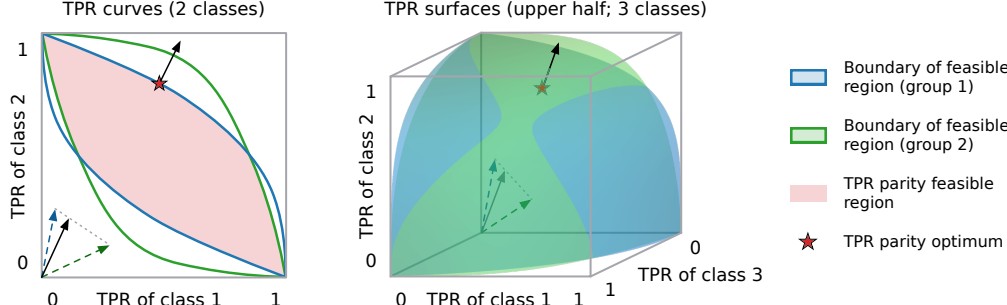

Figure 1: Feasible region of TPRs on a binary class (left) and a three-class problem (right). The black (resp. colored) arrow indicates the utility-maximizing direction (of each group).

regardless of demographics [20], e.g., being shown job postings on recruiting platforms for which the user is qualified. When the classes are binary, this fairness notion recovers *equalized odds* [23].

In this paper, we focus on the design of algorithm for mitigating TPR disparity and provide an efficient *post-processing* method that derives *attribute-aware* fair classifiers from (pre-trained) scoring models. Our method works on multi-class and multi-group classification problems, guarantees fairness by a sample complexity bound, can be implemented by linear programs, and achieves higher reductions in disparity compared to existing algorithms that are applicable to multi-class—a recently proposed post-processing method based on model projection [2], and adversarial debiasing [41], an in-processing method—especially when the number of classes is large.

**Organization.** We introduce the problem setup and objectives in Section 2, then describe our post-processing method for TPR parity in Section 3, along with suboptimality analyses; in particular, our method yields the optimal fair classifier when applied to the *Bayes optimal* score function. Our method is instantiated for finite sample estimation in Section 4, and we also provide sample complexity bounds to complete the analysis. Finally, in Section 5, we compare our algorithm with existing methods for disparity reduction on benchmark datasets.[1] A high-level summary of our results is provided in Section 1.1.

## 1.1 Summary of Results

One way to interpret and understand TPR parity is through visualizing the feasible regions of TPRs. In Fig. 1, we plot the feasible regions (achievable by probabilistic classifiers) of two groups on a (hypothetical) binary classification problem on the left, and those on a three-class problem on the right, where each axis represents the TPR of a class. Achieving optimal TPR parity amounts to first finding the TPR that maximizes the overall utility (e.g., accuracy) in the intersection of feasible regions, and subsequently an (attribute-aware) classifier attaining that target TPR on all groups. Note that the left figure is equivalent to the ROC curve (with a flip of the horizontal axis, because the TPR of class 1 equals one minus the false negative rate by treating class-1 as the negative class), which was used by Hardt et al. [23] for studying equalized odds. And thus, the TPR (hyper)surface plots in higher dimensions are a natural generalization of the ROC curve to multi-class settings.

Step one of finding the optimal fair TPR can be formulated as a linear program when estimating from finite samples. For the second step, our method derives a classifier attaining the target TPR from the score function; in particular, it yields the optimal fair classifier when the score is Bayes optimal:

**Theorem 1.1.** *Let $f_1^*, \cdots, f_m^* : \mathcal{X} \to \Delta_k$ denote the Bayes score function on each group, $f_a^*(x) \coloneqq \mathbb{E}[Y \mid X = x, A = a]$, and $q_1, \cdots, q_m \in \Delta_k$ be arbitrary. Then under a continuity assumption (2.3), $\exists \beta_1, \cdots, \beta_m \in [0,1]$ and $\lambda_1, \cdots, \lambda_m \in \mathbb{R}^k$ s.t. the probabilistic attribute-aware classifier*

$$(x, a) \mapsto \begin{cases} \arg\max_{y'} (\lambda_a)_{y'} \cdot f_a^*(x)_{y'} & \text{w.p. } 1 - \beta_a \tag{1a} \\ y & \text{w.p. } \beta_a \cdot (q_a)_y, \ \forall y \in [k] \tag{1b} \end{cases}$$

*achieves the maximum utility subject to TPR parity.*

---

[1]Our code is provided in the supplemental material.

The post-processed classifier returned by our method is a mixture of two models (weighted by $\beta$). Eq. (1a) returns the class with the highest likelihood after a class-wise rescaling, called a *tilting* [2], which generalizes the concept of *thresholding* in binary classifiers. Eq. (1b) makes random assignments sampled from a $\mathrm{Multinoulli}(q)$ distribution, which handles situations where the fair TPR lies in the interior of the feasible region (see Fig. 1, where the optimum is located within the interior of group 2 feasible region). To alleviate potential ethical concerns regarding this randomization, we point out that the parameter $q_a$'s used in class sampling can be specified per-group by the practitioner responsibly, e.g., uniform $\mathbf{1}/k$, or $e_{y'}$ with $y'$ being an advantaged outcome.

Among the possibly infinitely many fair classifiers derived from the score function $f$, our method specifically seeks the simplistic representation in Eq. (1) because it can be estimated via linear programs from finite samples. More importantly, it immediately extrapolates to unseen examples, and provides good generalization performance at the rate of $\widetilde{O}(\sqrt{k/n})$ thanks to its low function complexity (Theorem 4.2).

When the score function being post-processed is not Bayes optimal, our method is still applicable, but the resulting classifier may not be optimal nor exactly achieve TPR parity without access to labeled data (the method itself only needs unlabeled data with the sensitive attribute) or additional knowledge of the model. But these suboptimalities are minimized if the model is *calibrated* (Theorem 3.5); this answers the question raised in [2] about the effects of base model inaccuracies on downstream post-processing.

## 1.2 Related Work

**Fairness Critetia.** The notion of TPR parity has appeared in the literature as *conditional procedure accuracy equality* [7], *avoiding disparate mistreatment* [39], and (multi-class) *equal opportunity* [14, 29, 31] (to be distinguished from the fairness criterion with the same name in [23]). Other group fairness notions that extend to multi-class include (but not limited to) *equalized odds* [23] (of which TPR parity is a necessary condition), and *demographic parity* (DP) [10] (where Xian et al. [35] recently proposed an optimal post-processing method). However, DP may be less desirable than TPR parity in some use cases because the perfect classifier is not permitted under DP when the base rates differ [42]. It is worth noting that TPR parity implies *accuracy parity* [9]. In addition to group fairness, there are notions defined on the individual level [19].

**Mitigation Methods.** Our method is based on post-processing [25, 23]. There are also in-processing methods via fair representation learning [40, 41, 43, 30] or solving zero-sum games [1, 36], and pre-processing methods that debias the data prior to model training [11, 44]; see [4, 12] for a survey.

For multi-class TPR parity, the only applicable post-processing method to date, to our knowledge, is due to Alghamdi et al. [2] (which is the primary baseline for our method in our experiments). It is a general-purpose method that transforms the scores to satisfy fairness while minimizing the distributional divergence (e.g., KL) between the transformed scores and the original. However, the tradeoff between model performance and fairness is unclear as they did not relate the divergence to utility. Furthermore, while the authors provided a sample complexity bound for their optimization objective, it is not explicitly related to the violation of the fairness criteria.

## 2 Preliminaries

A $k$-class classification problem is defined by a joint distribution $\mu$ of input $X \in \mathcal{X}$, demographic group membership $A \in [m] := \{1, \cdots, m\}$ (a.k.a. the sensitive attribute), and class label $Y \in [k]$. We denote the joint distribution of $(X, A)$ by $\mu^{X,A}$, and, the $(k-1)$-dimensional probability simplex by $\Delta_k := \{z \in \mathbb{R}_{\geq 0}^k : \|z\|_1 = 1\}$.

Let $f : \mathcal{X} \times \mathcal{A} \to \Delta_k$ be an attribute-aware (pre-trained) score function, whose outputs are probability vectors that estimate the class probabilities as in $f(x, a)_y \approx \mathbb{P}_\mu(Y = y \mid X = x, A = a)$. We will write $f_a : \mathcal{X} \to \Delta_k$ to denote the component of $f$ associated with group $a$, i.e., $f_a(x) \equiv f(x, a)$. Our goal is to find fair (probabilistic) post-processing maps $g_1, \cdots, g_m : \Delta_k \to \mathcal{Y}$ to derive a classifier $(x, a) \mapsto g_a \circ f_a(x)$ that satisfies TPR parity while maximizing utility (e.g., classification accuracy).

We allow for controllable tradeoffs between utility and fairness through the following relaxation of TPR parity, and call a classifier $\alpha$-*fair* if it satisfies $\alpha$-TPR parity:

**Definition 2.1** (Approximate TPR Parity). Let $\alpha \in [0, 1]$. A predictor $\widehat{Y}$ is said to satisfy $\alpha$-TPR parity if $\Delta_{\mathrm{TPR}}(\widehat{Y}) \leq \alpha$, where

$$\Delta_{\mathrm{TPR}}(\widehat{Y}) \coloneqq \max_{a, a' \in \mathcal{A}} \left\| \mathrm{TPR}_a(\widehat{Y}) - \mathrm{TPR}_{a'}(\widehat{Y}) \right\|_\infty, \tag{2}$$

and $\mathrm{TPR}_a(\widehat{Y}) \coloneqq \mathbb{P}(\widehat{Y} \mid Y = y, A = a) \in [0, 1]^k$; $\mathbb{P}$ includes the randomness of the predictor.

Beyond classification accuracy, we also allow for any utility functions that depend only on the TPRs:[2]

**Definition 2.2** (Utility). The utility function $u : [k] \times [k] \to \mathbb{R}$ is defined for some $\upsilon \in \mathbb{R}^k$ by

$$u(\hat{y}, y) \coloneqq \sum_{y' \in [k]} \upsilon_{y'} \, \mathbb{1}[y = y', \hat{y} = y'].$$

E.g., accuracy, $\mathbb{1}[y = \hat{y}]$, is obtained by setting $\upsilon = \mathbf{1}_k$. The term $\upsilon$ will appear in our analyses, and the significance of considering utilities of this form is that we could evaluate a classifier by a weighted sum of its TPRs. Define $p_{ay} \coloneqq \mathbb{P}_\mu(A = a, Y = y)$, then

$$\mathcal{U}(\widehat{Y}) = \mathbb{E}\, u(\widehat{Y}, Y) = \sum_{a \in [m], y \in [k]} \upsilon_y p_{ay} \, \mathrm{TPR}_a(\widehat{Y})_y \equiv \mathcal{U}(\mathrm{TPR}_1(\widehat{Y}), \cdots, \mathrm{TPR}_m(\widehat{Y})). \tag{3}$$

Finally, we make the following continuity assumption on the distributions of score to avoid technical complexities related to tie-breaking (on the atoms). This assumption has also appeared in prior work on fair post-processing [16, 21, 35]; it holds when the input distributions are continuous and the score function is injective, or can be satisfied by adding small random perturbations to the scores.

**Assumption 2.3.** The conditional distribution of score, $\mathbb{P}(f_a(X) \mid A = a)$, is (Lebesgue absolutely) continuous, $\forall a \in [m]$.

## 3 TPR Parity via Post-Processing

Given a score function $f : \mathcal{X} \times \mathcal{A} \to \Delta_k$, and access to the (unlabeled) joint distribution $\mu^{X, A}$ (i.e., no estimation error), we describe a method for deriving an attribute-aware $\alpha$-fair classifier while maximizing utility, in the form of $(x, a) \mapsto g_a \circ f_a(x)$, where the $g_a$'s are (probabilistic) fair post-processing maps for each group. That is, we want to solve

$$\max_{g_1, \cdots, g_m} \mathcal{U}(\widehat{Y}) \quad \text{s.t.} \quad \Delta_{\mathrm{TPR}}(\widehat{Y}) \leq \alpha \quad \text{where} \quad \widehat{Y} = g_A \circ f_A(X).$$

Although the method only returns classifiers derived from $f$ as opposed to searching over the space of all classifiers $h : \mathcal{X} \times \mathcal{A} \to \mathcal{Y}$, it would yield the optimal fair classifier provided that the information of $(A, Y)$ is preserved in the output of $f$; this is the case when the score function is Bayes optimal.

### 3.1 Deriving Optimal Fair Classifier From Bayes Score Function

In this section, we explain how to obtain an optimal fair classifier by deriving from the Bayes score function $f^*$, thereby providing a *proof of Theorem 1.1* (omitted proof are deferred to the appendix).

**Step 1** (Finding Utility-Maximizing Fair TPRs). Let $D_a \subseteq [0, 1]^k$ denote the set of feasible TPRs on group $a$ achieved by probabilistic classifiers. The first step is to find utility-maximizing fair TPRs contained in an $\ell_\infty$-ball of diameter $\alpha$ per Definition 2.1 of $\alpha$-TPR parity (left figure of Fig. 2):

$$\max_{t_1 \in D_1, \cdots, t_m \in D_m} \mathcal{U}(t_1, \cdots, t_m) \quad \text{s.t.} \quad \|t_a - t_{a'}\|_\infty \leq \alpha, \ \forall a, a' \in [m]. \tag{4}$$

When $\alpha = 0$, this reduces to finding a single $t \in \bigcap_a D_a$, and because each $D_a$ is convex (since probabilistic classifiers are allowed), it can be found with ternary search as suggested in [23]. If instead the $t_a$'s are to be estimated from finite samples, then the empirical $\widehat{D}_a$'s are described by polytopes and the problem can be formulated as a linear program (Section 4).

---

[2]This includes all possible utility/loss functions in binary classification, since $\mathrm{TPR}(\widehat{Y})_1$ (true negative rate) and $\mathrm{TPR}(\widehat{Y})_2$ (true positive rate) fully determine the $2 \times 2$ confusion matrix.

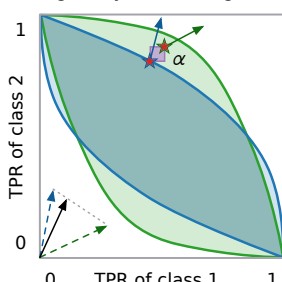
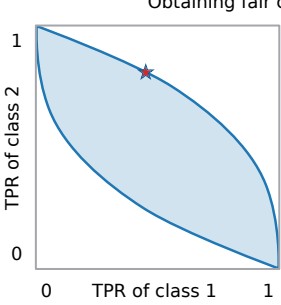
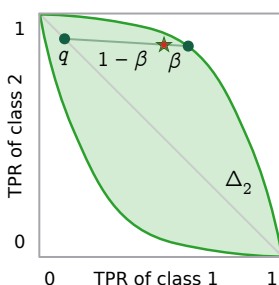

Step 1
Finding utility-maximizing fair TPRs

Step 2
Obtaining fair classifier of desired form

Figure 2: Achieving $\alpha$-TPR parity on a binary class problem. First, the utility-maximizing TPRs residing in an $\ell_\infty$-ball of diameter $\alpha$ are found (left). Then, classifiers achieving the fair TPRs are obtained: a tilting of the scores when the TPR lies on the boundary (middle), otherwise, a mixture of tilting and randomization (right). The simplex $\Delta_k$ is always inscribed in the feasible region.

The feasible regions of TPR generally differ across groups, due to uncertainties that are inherent to each group in the task of interest, or to inadequate and biased collection or sourcing of data. The more the $D_a$'s differ, the greater the tradeoff between fairness and utility; hence TPR parity incentivizes the practitioner to improve data collection and aspects of modeling that induces a balanced predictive capability on all groups [23].

Because $f^*(X, A)$ is *sufficient statistic* for $Y$, the fair TPRs we found above are always achievable by classifiers derived from $f^*$. Or more concretely,

**Proposition 3.1.** *Let $f^* : \mathcal{X} \to \Delta_k$ denote the Bayes score function, then $D := \{\mathrm{TPR}(h) \in [0, 1]^k \mid h : \mathcal{X} \to \mathcal{Y}$ (probabilistic)$\} = \{\mathrm{TPR}(g \circ f^*) \in [0, 1]^k \mid g : \Delta_k \to \mathcal{Y}$ (probabilistic)$\}$.*

**Step 2** (Obtaining Fair Classifier of Desired Form). Having found the utility-maximizing fair TPR $t_a$'s, the next step is to derive a classifier that attains $t_a$ on each group. This is provided by the following theorem:

**Theorem 3.2.** *Let $f^* : \mathcal{X} \to \Delta_k$ denote the Bayes score function, and $q \in \Delta_k$ be arbitrary. Then under Assumption 2.3, $\forall t \in D$, there exists $\beta \in [0, 1]$ and $\lambda \in \mathbb{R}^k$ s.t. $\mathrm{TPR}(h) = t$, where*

$$h(x) = \begin{cases} \arg\max_{y'} \lambda_{y'} f^*(x)_{y'} & \text{w.p. } 1 - \beta \\ y & \text{w.p. } \beta q_y, \ \forall y \in [k]. \end{cases}$$

The construction uses the observation that the boundary of $D$, denoted by $\partial D$, is given by the set of TPRs attained by tiltings of the Bayes score:

**Proposition 3.3.** *Let $f^* : \mathcal{X} \to \Delta_k$ denote the Bayes score function, then $h : \mathcal{X} \to \mathcal{Y}$ (probabilistic) satisfies $\mathrm{TPR}(h) \in \partial D$ if and only if $\exists \lambda \in \mathbb{R}^k$, $\lambda \neq 0$ s.t. $h(x) \in \arg\max_y \lambda_y f^*(x)_y$.*

*Proof of Theorem 3.2.* If the target TPR lies on the boundary of $D$, then by Proposition 3.3, it is achieved by a tilting of the Bayes score without any randomization (i.e., $\beta = 0$; center figure of Fig. 2). This holds due to Assumption 2.3, because we may break ties arbitrarily without affecting TPR, since the set of tied scores (finite union of $(k-2)$-d subspaces) has (Lebesgue) measure zero.

Otherwise, and generally, there must exists $t' \in \partial D$ and $\beta \in [0, 1]$ s.t. $t$ can be written as a linear combination of $t = \beta q + (1 - \beta)t'$. This is simply because $q \in \Delta_k \subseteq D$, and the line connecting $q$ and $t$ must intersect $\partial D$ at some point $t'$ (right figure of Fig. 2). Since the TPR of the input-agnostic randomization according to $\mathrm{Multinoulli}(q)$ equals $q$, and $t'$ is achieved by a tilting of the score per Proposition 3.3, their $\beta$-mixture achieves the target TPR $t$ by linearity. $\square$

## 3.2 Deriving From Any Score Function

The post-processing method described in the previous section, which only requires unlabeled data $(X, A)$, yields the optimal $\alpha$-fair classifier when applied to Bayes scores $f^*$. Yet, in practice, there

---

**Algorithm 1** Post-Process Score Function for $\alpha$-TPR parity

1: **Input:** $\alpha \in [0,1]$, $q_1, \cdots, q_m \in \Delta_k$, score function $f : \mathcal{X} \times \mathcal{A} \to \Delta_k$, distribution $\mu^{X,A}$
2: $\widetilde{D}_a := \{\widetilde{\mathrm{TPR}}_a(h) \mid h : \mathcal{X} \to \mathcal{Y} \text{ (probabilistic)}\}$     $\triangleright$ *Eq.* (5), *induced TPR feasible region*
3: $\tilde{t}_1, \cdots, \tilde{t}_m \leftarrow \arg\max_{\tilde{t}_1 \in \widetilde{D}_1, \cdots, \tilde{t}_m \in \widetilde{D}_m} \mathcal{U}(\tilde{t}_1, \cdots, \tilde{t}_m)$ s.t. $\|\tilde{t}_a - \tilde{t}_{a'}\|_\infty \leq \alpha$, $\forall a, a' \in [m]$
                      $\triangleright$ *utility-maximizing fair TPRs*
4: **for** $a = 1$ **to** $m$ **do**
5:   Find $h_a, \beta_a \in [0,1]$ s.t. $\widetilde{\mathrm{TPR}}_a(h_a) \in \partial \widetilde{D}_a$ and $\tilde{t}_a = (1 - \beta_a)\widetilde{\mathrm{TPR}}_a(h_a) + \beta_a q_a$
6:   Find $\lambda_a \in \mathbb{R}^k$ s.t. $h_a(x) \in \arg\max_{y'}(\lambda_a)_{y'} \cdot f_a(x)_{y'}, \forall x \in \mathrm{supp}(\mu_a^X)$
7: **end for**
8: **Return:** $(x,a) \mapsto \arg\max_{y'}(\lambda_a)_{y'} \cdot f_a(x)$ w.p. $1 - \beta_a$, and $y$ w.p. $\beta_a \cdot (q_a)_y$ for each $y \in [k]$

---

180   is the concern that Bayes score functions could be arbitrarily complex and are often not exactly
181   learnable due to limited data or computational constraints [34].

182   Nonetheless, our method is still applicable to arbitrary (approximations to the Bayes) score functions
183   $f : \mathcal{X} \times \mathcal{A} \to \Delta_k$ for deriving classifiers that are approximately fair and optimal, by treating them
184   *as if they were Bayes optimal* (Algorithm 1). Where, the only tweak we made is replacing the
185   ground-truth TPRs and feasible regions (which are unknown without access to the Bayes score) by
186   approximations *induced* by $f$, i.e.,

$$\widetilde{D}_a := \left\{\widetilde{\mathrm{TPR}}_a(h) \in [0,1]^k \,\middle|\, h : \mathcal{X} \to \mathcal{Y} \text{ (probabilistic)}\right\}, \tag{5}$$

187   where

$$\widetilde{\mathrm{TPR}}_a(h)_y := \frac{1}{\tilde{p}_{ay}} \int_{x \in \mathcal{X}} f_a(x)_y \, \mathbb{P}(h(x) = y) \, \mathrm{d}\mu^{X,A}(x,a), \quad \tilde{p}_{ay} := \int_{x \in \mathcal{X}} f_a(x)_y \, \mathrm{d}\mu^{X,A}(x,a). \tag{6}$$

188   It is not hard to show that they are equal to their ground-truth counterparts when $f = f^*$.

189   We may control and minimize the suboptimalities of the classifier returned from Algorithm 1 by
190   performing *group-wise distribution calibration* to the score function $f$ (using labeled data $(X, A, Y)$):

191   **Definition 3.4** (Distribution Calibration). *A score $R$ is said to be (group-wise) distribution calibrated*
192   *if $\mathbb{P}(Y = y \mid R = s) = s_y, \forall s \in \Delta_k, y \in [k]$ (resp. $\mathbb{P}(Y = y \mid R = s, A = a) = s_y, \forall a \in [m]$).*

193   Distribution calibration is a multi-class generalization of the original definition of calibration for
194   binary predictors [15, 32], requiring the predicted score to match the underlying class distribution
195   conditioned on the score across all classes, not just the most confident one [22]. Although this
196   definition is convenient to work with mathematically, it could be difficult to achieve in practice. In the
197   proof of Theorem 3.5, we relax it to a recently proposed notion of *decision calibration* [45] (w.r.t. the
198   set of all tiltings; derived from *multicalibration* [24]), which could be achieved in polynomial time.

199   **Theorem 3.5.** *Let $f : \mathcal{X} \times \mathcal{A} \to \Delta_k$ be a score function, and $h : \mathcal{X} \times \mathcal{A} \to \mathcal{Y}$ the (probabilistic)*
200   *classifier derived from $f$ using Algorithm 1. Then under Assumption 2.3, for any group-wise calibrated*
201   *reference score function $\bar{f} : \mathcal{X} \times \mathcal{A} \to \Delta_k$,*

$$\left|\overline{\mathcal{U}} - \mathcal{U}(h)\right| \leq \sum_{a \in [m], y \in [k]} 3\upsilon_y \epsilon_{ay}, \quad \Delta_{\mathrm{TPR}}(h) \leq \alpha + \max_{a \in [m], y \in [k]} \frac{4\epsilon_{ay}}{p_{ay}},$$

202   *where $p_{ay} := \mathbb{P}_\mu(A = a, Y = y)$, $\upsilon$ is from the utility function in Definition 2.2, $\overline{\mathcal{U}}$ denotes the utility*
203   *achieved by the optimal $\alpha$-fair classifier derived from the calibrated reference $\bar{f}$, and*

$$\epsilon_{ay} := \mathbb{E}\left[\left|\bar{f}_a(X)_y - f_a(X)_y\right| \cdot \mathbb{1}[A = a]\right]$$

204   *is the $L^1(\mu)$ difference between $f$ and the calibrated reference $\bar{f}$ on group $a$ and class $y$.*

205   We draw two conclusions from this result. First, by using the Bayes score function $f^*$ as the reference,
206   it states that the suboptimality of the derived classifier when $f \neq f^*$ is upper bounded by the
207   difference between the approximate scores and the ground-truth; this answers the question raised
208   in [2] regarding the impact of base model inaccuracies. Second, if $f$ satisfies calibration, then by
209   using itself as the reference, the result guarantees that the classifier derived using Algorithm 1 exactly
210   achieves the desired level of fairness, and is optimal among all fair classifiers derived from $f$ (which
211   cannot be improved without labeled data).

## 4 Finite-Sample Algorithm and Guarantees

We instantiate the post-processing method above for TPR parity to the case where we do not have access to the distribution $\mu^{X,A}$ but only samples drawn from it (i.e., to perform estimation), and analyze the sample complexity.

**Assumption 4.1.** We have $n$ i.i.d. (unlabeled) samples of $(X, A)$, which are independent of the score function $f$ being post-processed.

Denote the number of samples from group $a$ by $n_a$, and the samples themselves by $(x_{a,i})_{i \in [n_a]}$.

### 4.1 Algorithm

We adapt Algorithm 1 to handle finite samples by replacing $\widetilde{D}_a$ and $\mathcal{U}$ with their empirical counterparts (essentially calling it with the empirical distribution $\hat{\mu}^{X,A}$ formed by the samples as the argument), and implement the optimization problems on Lines 3, 5 and 6 using linear programs.

**Step 1** (Finding Utility-Maximizing Fair TPRs). The empirical induced feasible region of TPRs, $\widehat{D}_a$, can be computed via evaluating the TPRs of all (probabilistic) classifiers acting on the samples—by representing them using $n_a \times k$ lookup tables (each row gives the probabilities of the random class assignment on the corresponding sample):

$$\widehat{D}_a := \left\{ \widehat{\mathrm{TPR}}_a(\gamma_a) \;\middle|\; \gamma_a \in \mathbb{R}_{\geq 0}^{n_a \times k}, \; \textstyle\sum_{y \in [k]} (\gamma_a)_{i,y} = 1, \; \forall i \in [n_a] \right\},$$

where

$$\widehat{\mathrm{TPR}}_a(\gamma)_y := \frac{1}{n\hat{p}_{ay}} \sum_{i \in [n_a]} f_a(x_{a,i})_y \cdot (\gamma_a)_{i,y}, \quad \hat{p}_{ay} := \frac{1}{n} \sum_{i \in [n_a]} f_a(x_{a,i})_y$$

(cf. Line 2 and Eqs. (5) and (6)). Note that $\widehat{D}_a$ is a polygon, as it is specified by linear constraints.

To obtain the utility-maximizing fair TPR $\hat{t}_a$'s, we take the empirical maximizer subject to the $\alpha$-TPR constraint via solving a linear program (cf. Line 3 and Eqs. (3) and (4)):

$$\mathrm{LP1}(\alpha): \max_{\hat{t}_1 \in \widehat{D}_1, \cdots, \hat{t}_m \in \widehat{D}_m} \widehat{\mathcal{U}}(\hat{t}_1, \cdots, \hat{t}_m) \quad \text{s.t.} \quad \|\hat{t}_a - \hat{t}_{a'}\|_\infty \leq \alpha, \; \forall a, a' \in [m],$$

where $\widehat{\mathcal{U}}(\hat{t}_1, \cdots, \hat{t}_m) := \sum_{a,y} v_y \hat{p}_{ay} (\hat{t}_a)_y$ is the empirical utility.

**Step 2** (Obtaining Fair Classifier of Desired Form). The next step is finding a classifier that achieves $\hat{t}_a$'s on the empirical distribution, i.e., Lines 5 and 6. To implement Line 5, note that another way of approaching this problem is to realize that among all eligible $(\beta_a, h_a)$-pairs, the $h_a$ associated with the maximum $\beta_a$ value must satisfy $\widetilde{\mathrm{TPR}}_a(h_a) \in \partial \widetilde{D}_a$ (otherwise, a contradiction can be reached using the fact that $\widetilde{D}_a \subseteq [0,1]^k$ is compact; also see the right figure of Fig. 2). Combined with the strategy above of representing classifiers using lookup tables, we get the following linear program:

$$\mathrm{LP2}(t, q): \max_{\gamma, \beta} \beta \quad \text{s.t.} \quad t = (1 - \beta)\widehat{\mathrm{TPR}}(\gamma) + \beta q \quad \text{and} \quad \gamma \in \mathbb{R}_{\geq 0}^{n \times k}, \; \sum_{y \in [k]} \gamma_{i,y} = 1, \; \forall i \in [n].$$

Finally, on Line 6, we find a tilting $\lambda_a$ s.t. after coordinate-wise multiplied by the scores, the argmax class assignment has nonzero probability according to the classifier $\gamma_a$ found in the preceding step:

$$\mathrm{LP3}(\gamma): \min_\lambda 0 \quad \text{s.t.} \quad \lambda_y f(x_i)_y \geq \lambda_{y'} f(x_i)_{y'} \quad \forall i \in [n], \; y, y' \in [k], \; \gamma_{i,y} > 0.$$

The feasible set of this problem is nonempty by Proposition 3.1, because we are *treating $f$ as if it were the Bayes score function, and the empirical distribution $\hat{\mu}^{X,A}$ as the population.*

All combined, our algorithm involves solving $(2m + 1)$ linear programs, where LP1 is the dominating one with $O(nk)$ variables and constraints; solving which (to near-optimality) takes, e.g., $\widetilde{O}(\mathrm{poly}(nk))$ time using interior point methods [33].

## 4.2 Sample Complexity

Thanks to the low function complexity of post-processing maps used in our algorithm to derive classifiers (Eq. (1)), it enjoys the following efficient sample complexity:

**Theorem 4.2.** *Let $f : \mathcal{X} \times \mathcal{A} \to \Delta_k$ be a score function, and $h : \mathcal{X} \times \mathcal{A} \to \mathcal{Y}$ the (probabilistic) classifier derived from $f$ using Algorithm 1 with the empirical distribution formed by samples from Assumption 4.1 as the argument. Then under Assumption 2.3, for any group-wise calibrated (Definition 3.4) reference score function $\bar{f} : \mathcal{X} \times \mathcal{A} \to \Delta_k$, and $n \geq \Omega(\max_{a,y} \ln(mk/\delta)/p_{ay})$,*

$$\left| \overline{\mathcal{U}} - \mathcal{U}(h) \right| \leq O\left( \sum_{a \in [m], y \in [k]} v_y \left( \sqrt{\frac{k p_{ay}}{n} \ln \frac{mk}{\delta}} + \frac{k}{n} + \epsilon_{ay} \right) \right),$$

$$\Delta_{\mathrm{TPR}}(h) \leq \alpha + O\left( \max_{a \in [m], y \in [k]} \left( \sqrt{\frac{k}{n p_{ay}} \ln \frac{mk}{\delta}} + \frac{k}{n p_{ay}} + \frac{\epsilon_{ay}}{p_{ay}} \right) \right),$$

*where $\overline{\mathcal{U}}$ denotes the utility achieved by the optimal $\alpha$-fair classifier derived from the calibrated reference $\bar{f}$, and $\epsilon_{ay} := \mathbb{E}[|\bar{f}_a(X)_y - f_a(X)_y| \cdot \mathbb{1}[A = a]]$.*

The bound consists of a calibration error $\epsilon_{ay}$ as discussed in the remarks of Theorem 3.5, an estimation error from applying uniform convergence (the Natarajan dimension of the set of tiltings is $O(k)$), and a $k/n$ term that comes from the disagreement over class assignments on the samples between the (deterministic) tilting found on Line 6 and the (probabilistic) classifier on Line 5 due to tie-breaking.

## 5 Experiments

We evaluate Algorithm 1 for reducing TPR disparity on benchmark datasets, and demonstrate its effectiveness compared to existing post-processing as well as in-processing bias mitigation methods.

**Datasets.** The first task is income prediction, for which, we use the `ACSIncome` dataset [18]—an extension of the UCI Adult dataset [27] with much more examples (1.6 million vs. 30,162), allowing us to compare methods confidently. We consider a binary setting where the sensitive attribute is gender and the target is whether the income is over $50k, as well as a multi-group multi-class setting with five race categories and five income buckets. The second is text classification, of identifying occupations (28 in total) from biographies in the `BiasBios` dataset [14]; sensitive attribute is gender.

**Baselines and Setup.** The main baseline is `FairProjection` [2]—the only post-processing algorithm applicable for multi-class TPR parity to our knowledge.[3] In the binary setting, we also compare to `RejectOption` [25]. To demonstrate the deficiencies of existing methods at reducing TPR disparity, we additionally include in-processing results using `Reductions` [1] and `Adversarial` [41].[4][5]

On each task, we first create a pre-training split from the dataset and train a linear logistic regression scoring model (with isotonic calibration and five-fold cross-validation as implemented in `scikit-learn` [37, 38, 28]), then randomly split the remaining data for post-processing and testing with 10 different seeds and aggregate the results (the pre-trained model remains the same). For in-processing, we use the same splits but merge the pre-training and post-processing data for training. On `BiasBios`, linear logistic regression is performed on the embeddings of the biographies computed by a previously fine-tuned BERT model [17] (in other words, head-tuning). Additional details including hyperparameters are included in the appendix.

**Results.** In Fig. 3, we plot the tradeoff curves from varying the fairness tolerance ($\alpha$ for our method). Our method is consistently the most effective at minimizing TPR disparity, particularly under multi-class settings, where existing algorithms only manage to partially reduce $\Delta_{\mathrm{TPR}}$ (and at a greater cost to accuracy when using `FairProject` and `RejectOption`). It also outperforms

---

[3]We use the authors' code, where TPR parity is equivalent to the `meo` constraint. The results from using the KL divergence variant is included, which are better than the cross-entropy variant in our experiments.

[4]Although `Reductions` is extended to multi-class by Yang et al. [36], an implementation was not provided.

[5]The implementation (with minor modifications) in the `AIF360` library is used for the latter methods [5].

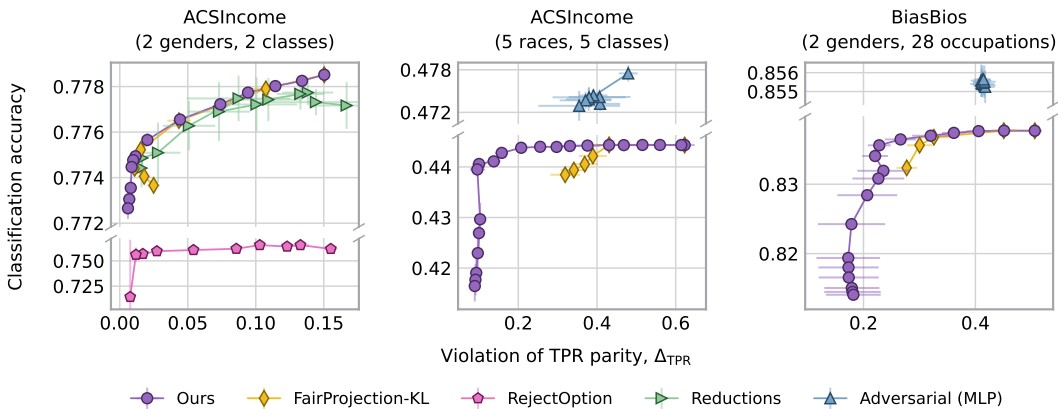

Figure 3: Tradeoff curves between accuracy and $\Delta_{\text{TPR}}$ (Eq. (2)). The base model is logistic regression (except for Adversarial, which uses a feedforward network). Error bars indicate the standard deviation over 10 runs with different random dataset splits. Running time is reported in the appendix.

the in-processing `Reductions` on binary `ACSIncome`, and `Adversarial` in terms of $\Delta_{\text{TPR}}$, which, although enjoys higher accuracies because of the use of the more expressive feedforward networks as the prediction model, fails to reduce TPR parity. Sharper drops in accuracies are observed when applying our method with small $\alpha$ settings, e.g., $0.001$ to $0.0001$. We saw this happen when the randomized component in Eq. (1b) is activated (i.e., $\beta > 0$), meaning that Line 3 finds a fair TPR that lies in the interior of the feasible region of the better-performing group in order to match the feasible TPR on the worse-performing one(s). Hence the drop is expected because utility is being sacrificed to achieve TPR parity.

Although our method greatly reduces TPR disparity, there remains a gap to reaching $\Delta_{\text{TPR}} = 0$, especially on tasks with more classes (i.e., `BiasBios`, where a higher variance is also observed). While this could be due to miscalibration, or potentially a violation of Assumption 2.3, the main reason is suspected to be insufficient sample size. Recall from Theorem 4.2 that the sample complexity for $\Delta_{\text{TPR}}$ scales as $\widetilde{O}(\sqrt{k/np_{ay}})$ in the worse-case $(a, y)$, which is itself at least $\widetilde{O}(\sqrt{mk^2/n})$. Thus, learning generalizable classifiers that satisfy TPR parity under more groups and classes is much harder in terms of data requirement (and by extension, computing resource).

Lastly, we emphasize the necessity of group-wise calibration for achieving low $\Delta_{\text{TPR}}$, as the definition of the criterion involves conditioning on the true label (it is also reflected by the calibration error term $\epsilon_{ay}$ in Theorem 4.2). In an ablation study in the appendix, a larger (minimum achievable) $\Delta_{\text{TPR}}$ is observed when no efforts are made to calibrate the scoring model. It is therefore necessary for model vendors to provide accurate uncertainty quantifications, and for practitioners building fair classifiers to verify and improve calibration.

## 6 Conclusions and Limitations

We described a post-processing method for reducing TPR disparity for equal opportunity in multi-class classification, and demonstrated its performance in comparison to existing algorithms on benchmarks datasets, especially when the number of classes is large. We analyzed the sample complexity of our method, and established its optimality under model calibration.

The effectiveness of our method at reducing TPR disparity is largely contributed to the tailored analysis, although it limits our method to this fairness notion only. Some use cases may demand equalized odds ($\widehat{Y} \perp A \mid Y$) beyond TPR parity ($\mathbb{1}[\widehat{Y} = Y] \perp A \mid Y$), which is a more stringent criterion: TPR parity only needs to match the main diagonal of the (conditional) confusion matrix across groups, whereas equalized odds requires matching all $k^2$ entries. The design of efficient algorithms for achieving equalized odds remains an open problem.[6]

---

[6]We note that most (general-purpose) fairness algorithms, e.g., [2], are only evaluated for TPR parity but not equalized odds.

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
