# OpenReview forum: "Efficient Post-Processing for Equal Opportunity in Fair Multi-Class Classification"
_NeurIPS.cc/2023/Conference — Submitted to NeurIPS 2023_

### Official Review · Reviewer_Hx2f · 2023-06-08

**Soundness:** 3 good
**Presentation:** 2 fair
**Contribution:** 1 poor
**Rating:** 3
**Confidence:** 4

**Summary:**

This paper considers fairness in multi-class classification under the notion of parity of true positive rates - an extension of binary class equalized odds - which ensures equal opportunity to qualified individuals regardless of their demographics. We focus on algorithm design and provide a post-processing method that derives fair classifiers from pretrained score functions.

**Strengths:**

- paper deals with an interesting problem
- paper is technically sound

**Weaknesses:**

- paper does not properly compare w.r.t. the state of the art
- novelty of the proposal is not clear
- paper is hard to read and follow

**Questions:**

Paper does not properly compare w.r.t. the state of the art, there are plenty of work (even in NeurIPS) that are not even mentioned in the paper, why these work are not mentioned?
Novelty of the proposal is not clear because of a proper state-of-the-art review and then it is not clear the contribution. Can you please elaborate and descrive the technical challenges of the contribution?

**Limitations:**

Paper is very hard to read, follow, and fully understand.

---

> ### Author Rebuttal · Authors · 2023-08-10
>
> We thank the reviewer for the time, comments, and support! We hope that the following addresses the concerns:
>
> - **(W2, Q3) "novelty of the proposal is not clear... elaborate and descrive the technical challenges of the contribution?"**
>
> 	We would like to kindly remind the reviewer of our main contributions:
>
> 	1. Identifying the representation of the optimal fair classifier under TPR parity. Most importantly, it has low function complexity.
> 	2. Providing a post-processing algorithm for achieving TPR parity.  Crucially, we provide generalization as well as optimality guarantees.
> 	3. The method works on multi-class, multi-group problems.
>
> 	Most prior work in fairness has focused on binary classification and univariate regression problems, however, with few exceptions, most of them cannot be extended to multi-class problems.  We also refer the reviewer to [2, table 1] for a review on prior work on fairness, which shows that most existing methods are inapplicable to multi-class problems.
>
> 	Below, we list the technical barriers that prohibits trivial extensions of methods for binary classification to multi-class:
>
> 	1. Most methods (e.g., [1, 2]) involve adjusting the threshold of the univariate score function to achieve output parity.  But in multi-class problems, the score function is multivariate, and it is not clear what should be the multi-class equivalent for thresholding.  In our work, we show that the appropriate analogue for TPR parity is "tilting".
> 	2. Some work approached problem (e.g., [3, 4]) by computing the Lagrangian of the constrained optimization problem, then deriving expressions for the fair classifier.  However, their derivations leveraged the assumption of binary class or binary group.
> 	3. In terms of statistical guarantees, without a careful design, a naive extension will have sample complexity that scales exponentially with the number of dimension or classes $k$ (whereas ours is linear in $k$).
>
> 	Moreover, most prior work (even in binary/univariate settings) do not provide optimality analysis nor statistical guarantees (our Theorems 4.2 and 3.5).
>
> - **(W1, Q1, Q2) "a proper state-of-the-art review [is missing]" and "paper does not properly compare w.r.t. the state of the art"**
>
> 	We summarized the SOTA methods that are **applicable to (multi-class) equal opportunity** in Section 1.2, lines 96–105.  The methods are also compared to in our experiments:
>
> 	+ RejectOption (only applicable to binary classification) [6].
> 	+ Reductions (only applicable to binary classification) [7].
> 	+ Adversarial debiasing [8].
> 	+ FairProjection [5].  This is the most recent SOTA fairness algorithm for multi-class problems, and we share the same baseline methods in our experiments with theirs.
>
> 	Could the reviewer elaborate on the SOTA approaches, applicable to our setting, that we are missing?
>
> - **(W3, Limitations) "paper is hard to read and follow"**
>
> 	We provided a high-level summary of our main results and methodology in Section 1.1, and structured the technical discussions in the subsequent sections following the flow in Section 1.1. We also included figures for illustrating the steps taken by our algorithm in Fig. 2, and clearly listed the algorthm in Algorithm 1.
>
> 	Could the reviewer kindly elaborate on which parts of the paper is unclear, and in which aspects?
>
> [1] Equality of Opportunity in Supervised Learning. NeurIPS 2016.
> [2] Bayes-Optimal Classifiers under Group Fairness. 2022.
> [3] Fair learning with Wasserstein barycenters for non-decomposable performance measures.  AISTATS 2023.
> [4] Fairness guarantee in multi-class classification. 2023.
> [5] Beyond Adult and COMPAS: Fair Multi-Class Prediction via Information Projection.  NeurIPS 2022.
> [6] Decision Theory for Discrimination-Aware Classification. ICDM 2012.
> [7] A Reductions Approach to Fair Classification. ICML 2018.
> [8] Mitigating Unwanted Biases with Adversarial Learning.  AAAI 2018.

---

> > ### Comment · Reviewer_Hx2f · 2023-08-15
> >
> > Many important works are not mentioned e.g. [1] and [2], together with many others in NeurIPS.
> > I will keep my score as it is.
> >
> > [1] Chzhen, E. et al. Leveraging Labeled and Unlabeled Data for Consistent Fair Binary Classification, NeurIPS 2019.
> > [2] Ray, J. et al. Wasserstein fair classification, UAI, 2020.

---

> > > ### Author Response · Authors · 2023-08-15
> > > **Thank you reviewer Hx2f**
> > >
> > > We would like to thank the reviewer for the response!
> > >
> > > - We have run additional evaluation comparing to [1], which will be included in our revision:
> > >
> > > 	| ACSIncome | accuracy | $\Delta_\text{TPR}$ |
> > > 	| --- | --- | --- |
> > > 	| Ours (tolerance = 0.0001) | 0.7727 ± 0.0005 | 0.0060 ± 0.0031 |
> > > 	| Ours (tolerance = 0.005) | 0.7745 ± 0.0003 | 0.0087 ± 0.0024 |
> > > 	| Ours (tolerance = 0.008) | 0.7748 ± 0.0003 | 0.0099 ± 0.0013 |
> > > 	| LevEqOpp | 0.7748 ± 0.0004 | 0.0088 ± 0.0016 |
> > >
> > > - We would like to point out that [2] is for demographic parity, not the equal opportunity considered in our work, but we would be happy to include the references provided by the reviewer in our revision.  Comparisons to many existing applicable methods are made in our experiments as presented in Fig. 3.

---

### Official Review · Reviewer_TMc5 · 2023-06-12

**Soundness:** 3 good
**Presentation:** 3 good
**Contribution:** 2 fair
**Rating:** 6
**Confidence:** 4

**Summary:**

This paper proposes a novel post-processing approach to reduce the true positive rate parity for multi-class classification problems. It is shown on two real world data to outperform an existing baseline in terms of accuracy and true positive rate parity.

**Strengths:**

1. The proposed approach is novel and technically sound.

2. The proposed approach guarantees fairness by a sample complexity bound.

3. The proposed approach outperforms an existing baseline post-processing approach in terms of reducing TPR parity.

4. The presentation is clear.

**Weaknesses:**

1. Some related work, such as [45], is mentioned in the paper but not compared in the experiments.

2. Fairness in multi-class problems seems to be a rather trivial/incremental problem when there are plenty of approaches solving the fairness problems in binary classification and regression problems. Approaches from fairness in regression (e.g. Narasimhan et al., "Pairwise fairness for ranking and regression", 2020) should be easily applied to solve the fairness problems in multi-class classification. Please justify more for why Fairness in multi-class classification is a problem worth studying. Some approaches from fairness in binary-classification (e.g. Kamiran and Calders, "Data preprocessing techniques for classification without discrimination", 2012; Yu et al., "FairBalance: How to Achieve Equalized Odds With Data Pre-processing", 2023) can also be easily adapted to multi-class problems.

**Questions:**

My main concern is about the significance and contribution of fairness in multi-class classification. To me, this is a relatively trivial problem compared to fairness in binary-classification and regression. Many approaches from fairness in binary-classification and regression can be easily adapted to solve this problem.

Q1. Why not compare this approach to [45]?

Q2. Why is fairness in multi-class classification is an important problem? Can existing work from fairness in binary-classification or regression be applied to solve this problem?

---

> ### Author Rebuttal · Authors · 2023-08-10
>
> We thank the reviewer for the time, comments, and support! We hope that the following addresses the concerns:
>
> - **(Q1) "Why not compare this approach to ["Learning Fair Representations"]?"**
>
> 	The group fairness criterion considered in [1] is for statistical parity, $A\perp Y$, not equal opportunity (which was not proposed by [2] until 2016), hence not comparable to our method (note that the two notions are generally incompatible, see e.g., [3]).
>
> 	But, the idea of [1] is to learn a fair feature representation $Z$ s.t. $Z\perp A$, and it has been extended to equalized odds ($Z\perp A\mid Y$ $which implies TPR parity) by [4] via adversarial debiasing.  Our experiments includes [4] (Fig. 3).
>
> - **(W2, Q2) "Fairness in multi-class problems seems to be a rather trivial/incremental problem when there are plenty of approaches solving the fairness problems in binary classification and regression problems"**
>
>
> 	Most prior work in fairness has focused on binary classification and univariate regression problems, however, we would like to point out that, with few exceptions, most of them cannot be extended to multi-class problems.  We also refer the reviewer to [5, table 1] for a review on prior work on fairness, which shows that most existing methods are inapplicable to multi-class problems.
>
> 	Below, we list the technical barriers that prohibit trivial extensions of methods for binary classification to multi-class:
>
> 	1. Most methods (e.g., [2, 8]) involve adjusting the threshold of the univariate score function to achieve output parity.  But in multi-class problems, the score function is multivariate, and it is not clear what should be the multi-class equivalent for thresholding.  In our work, we show that the appropriate analogue for TPR parity is "tilting".
> 	2. Some work approached problem (e.g., [6, 7]) by computing the Lagrangian of the constrained optimization problem, then deriving expressions for the fair classifier.  However, their derivations leveraged the assumption of binary class or binary group.
> 	3. In terms of statistical guarantees, without a careful design, a naive extension will have sample complexity that scales exponentially with the number of dimension or classes $k$ (whereas ours is linear in $k$).
>
> 	Moreover, our main contribution are:
>
> 	1. Identifying the representation of the optimal fair classifier under TPR parity. Most importantly, it has low function complexity.
> 	2. Providing a post-processing algorithm for achieving TPR parity.  Crucially, we provide generalization as well as optimality guarantees.
>
> 	Yet, most prior work (even in binary/univariate settings) do not provide optimality analysis nor statistical guarantees (our Theorems 4.2 and 3.5).
>
> - **(W2, Q2) "(e.g. Narasimhan et al., 'Pairwise fairness for ranking and regression', 2020) should be easily applied to solve the fairness problems in multi-class classification".**
>
> 	This paper considers ranking and regression, and they did not mention whether their method is related to classification.  It is not clear how it can be "easily applied to solve the fairness problems in multi-class classification".  Moreover, if we were to consider multivariate regression, we do not immediately see how their method can be extended, since they require being able to compare the value of two scores ($y< y'$ or $y>y'$), but in multivariate settings, the scores are multi-dimensional (how to compare two vectors?).
>
> 	In addition, our results have generalization as well as optimality guarantees (Theorems 4.2 and 3.5).  Such results are absent in their paper, so even if we were able to extend it to (multi-class) classification, we will not get similar guarantees in our paper.
>
> - **(W2, Q2) "Some [pre-processing] approaches [...] can also be easily adapted to multi-class problems."**
>
> 	Indeed, pre-processing approaches can be easilt adapted to multi-class problems, but they do not come with generalization nor optimality guarantees.  Most crucially, our method provides statistical guarantees on **fairness** when applied on the population.  In contrast, pre-processing methods perform subsampling and reweighting to balance the training set, but they do not provide guarantees that models trained on their dataset will be fair.  In this regard, extending existing pre-processing methods to multi-class will not have the guarantees in our paper.
>
> [1] Learning Fair Representations.  ICML 2013.
> [2] Equality of Opportunity in Supervised Learning. NeurIPS 2016.
> [3] Fairness and Machine Learning: Limitations and Opportunities. MIT Press, 2023.
> [4] Mitigating Unwanted Biases with Adversarial Learning.  AAAI 2018.
> [5] Beyond Adult and COMPAS: Fair Multi-Class Prediction via Information Projection.  NeurIPS 2022.
> [6] Fair learning with Wasserstein barycenters for non-decomposable performance measures.  AISTATS 2023.
> [7] Fairness guarantee in multi-class classification. 2023.
> [8] Bayes-Optimal Classifiers under Group Fairness. 2022.

---

> > ### Comment · Reviewer_TMc5 · 2023-08-15
> >
> > I agree with some of the clarifications the authors made regarding the difference between multi-class classification and regression problems. Hence my rating has been raised. However, the contribution is still limited for just solving multi-class classification problems. Contribution will be larger if the authors could compare the approach with other existing ones adapted to multi-class classification problems or showing that the proposed approach has state-of-the-art performances on other settings (binary classification, regression, etc.).

---

> > > ### Author Response · Authors · 2023-08-15
> > > **Thank you reviewer TMc5**
> > >
> > > We thank the reviewer for their time and support!
> > >
> > > - Regarding binary classification setting, we would like to point out that our method applies to all classification problems, including binary classification. The left panel of Figure 3 shows the results on a binary classification problem, where our method can match, if not better than, the performance of a variety of existing methods. In addition, our method is also mostly on-par with [1] (please kindly find the results in our response to area chair 8jVU above), which area chair 8jVU commented that "in the benchmarks provided in Alghamdi et al. [2], [it] seems to perform very well".
> > >
> > > - Regarding comparing to existing approaches for binary classification adapted to multiclass, we are not aware of an obvious way of easily adapting existing methods for binary classification to multiclass, but nontrivial analyses and reformulation are typically required for such extensions (e.g., [2] extends [3], and [4] extends [5] but they did not include multiclass experiments/code). We provide code for reproducing our results, so that future work on extending existing methods or on entirely new methods can compare with our method.
> > >
> > > [1] Leveraging Labeled and Unlabeled Data for Consistent Fair Binary Classification. NeurIPS 2019.
> > > [2] Beyond Adult and COMPAS: Fair Multi-Class Prediction via Information Projection. NeurIPS 2022.
> > > [3] Optimized Score Transformation for Consistent Fair Classification. JMLR 2021.
> > > [4] Fairness with Overlapping Groups; a Probabilistic Perspective. NeurIPS 2020.
> > > [5] A Reductions Approach to Fair Classification. ICML 2018.

---

### Official Review · Reviewer_EVhV · 2023-07-07

**Soundness:** 4 excellent
**Presentation:** 3 good
**Contribution:** 3 good
**Rating:** 6
**Confidence:** 4

**Summary:**

The work proposes a post-processing algorithm to achieve the equal opportunity constraint in multi class classification. The proposed algorithm takes arbitrary Bayes rule estimate and only requires additional unlabelled data. The authors derive finite-sample guarantees and perform empirical evaluation to support their claims.

I did not check the math, but, having prior experience in this area it looks believable.

**Strengths:**

The paper is well written and the proposed methodology is sound. It extends a rather long line of works on post-processing with unlabelled data.

**Weaknesses:**

In the context of the paper, I do not see major weaknesses from the methodological side, but rather remarks that are presented in the next part.

From the theoretical part, I could mention that the devision by p_{a, y} with large number of protected attribute and classes can make the bound non-informative.

I think that the expectation in the fairness guarantee is not well placed, I expected to have  $E[\Delta(h)]$.

**Questions:**

1. Could the authors formulate a theorem, stating explicit form of the optimal fair classifier in their setup?
2. In the very end of conclusion, the authors say that designing post-processing algorithm for equal odds remain open. While formally correct, isn't it true that one can do exactly the same analysis?
3. While assumption 2.3 did appear in many previous works, all of them were dealing purely with deterministic classifiers (regressors). Actually, the only purpose of this assumption is to avoid randomization on some tie-breaks. Since you still do randomise, why do you need this assumption?
4. Many works explicitly present the form of the optimal fair classifier (e.g., [16, 21, A, B] ...), which is not the case here (it is more opaque here). Could you sate the result unpacking the exact form?
5. What's the issue to generalise it to attribute unaware case? One only needs to perform a post-processing on top of the Bayes rule and A | X, Y or A | X.
6. Could the authors discuss computation complexity of their method, explicating the dependency on all the parameters?

To the best of my knowledge the first use of unlabelled data for post-processing in binary classification with equal opportunity was done in [A]. In [A], authors only consider the binary case and exact fairness, while targeting the optimal fair classifier.


[A] E. Chzhen, C. Denis, M. Hebiri, L. Oneto, M. Pontil, Leveraging Labeled and Unlabeled Data for Consistent Fair Binary Classification, 2019
[B] N. Schreuder, E. Chzhen, Classification with abstention but without disparities


**Limitations:**

---

---

> ### Author Rebuttal · Authors · 2023-08-10
>
> We thank the reviewer for the time, detailed comments, and support! We hope that the following addresses the concerns:
>
> - **(Q1, Q4) "explicitly present the form of the optimal fair classifier"**
>
> 	Thank you for the suggestion! We provide below the explicit expression (although not closed-form, which is typical, see e.g., [B, Theorem 3.2]) for our optimal fair classifier (after putting everything together), which will be included in our revision:
>
> 	> Let $f^*_1,\cdots,f^*_m$ denote the Bayes score function on each group, then the probabilistic attribute-aware classifier achieving the maximum utility subject to TPR parity takes the form of
> 	>
> 	> $$(x,a)\mapsto \begin{cases} g_{1,a}(x) & \text{w.p. } 1-\beta_a\\\\ g_{2,a}(x) & \text{w.p. } \beta_a,  \end{cases}$$
> 	>
> 	> where, for any arbitrary $q_a\in\Delta_k$, we define
> 	>
> 	> $$\begin{aligned}\text{(deterministic)}\quad g_{1,a}(x) &= \textstyle{\arg\max_{y'}} (\lambda_a)_{y'}   f^*_a(x) _{y'}, \\\\ \text{(randomized)} \quad g _{2,a}(x) &= y \text{ w.p. }  \beta_a (q_a) _y, \\,\forall y\in[k], \end{aligned}$$
> 	>
> 	> with $(\beta_a, \lambda_a)$ a feasible solution to the following problem (LP2 and LP3 in our algorithm; existence guaranteed by Proposition 3.3 and Theorem 3.2):
> 	>
> 	>$$\text{find } \beta _a\in[0,1], \lambda _a\in \mathbb R^k  \text{ s.t. }  t_a = (1-\beta_a) \, \mathrm{TPR} _a(g _{1,a} )  + \beta _a \, \mathrm{TPR} _a(g _{2,a}),$$
> 	>
> 	>and $t_a$ the utility-maximizing fair TPR (our LP1):
> 	>
> 	> $$\max_{t_1\in D_1,\cdots, t_m\in D_m} \mathcal U(t_1,\cdots,t_m) \text{ s.t. } \|t_a-t_{a'}\|_\infty\leq \alpha,\, \forall a,a'\in[m].$$
>
> - **(Q2) "for equal odds[...] isn't it true that one can do exactly the same analysis"**
>
> 	Thank you for raising this question!  First, we would like to point out that the generalization guarantee and good performance of our algorithm are enabled by the existence of the optimal TPR-fair classifier in the simplistic form presented above (a tilting + a randomized function).   Despite our best effort working on this project (indeed, our initial goal was to tackle equalized odds), we cannot find a similarly simple expression for the optimal EqOdds-fair classifier.
>
> 	In particular, our analysis is mainly based on the TPR curves/surfaces (Fig. 1), and the observation that any TPR on the boundary can be achieving by a tilting (Proposition 3.3).  But this cannot be extended for studying EqOdds because TPR curves/surfaces do not include information of the off-diagonal elements of the row-normalized confusion matrix (note that the elements of the diagonals are the TPRs), which are needed for EqOdds since it requires all entries to match.  We will include this discussion in our revision.
>
> - **(Q3) "why do [we] need [assumption 2.3]?"**
>
> 	First, recall from above that our optimal fair classifier has the form of a tilting $g_1$ + an input-agnostic randomized function $g_2$. Similar to prior work, we use Assumption 2.3 to deal with tie-breaking when using the tilting. We still require randomization in two places: (1) $\beta$ for interpolating between $g_1$ and $g_2$, and (2) in the definition of $g_2$; it is not for tie-breaking, but for achieving the fair TPRs found in Eq. (4).
>
> 	When the fair TPR lies on the boundary of the feasible region (figure 2 middle), proposition 3.3 tells us that we can achieve it with a tilting (which is deterministic).  But when the fair TPR lies in the interior (figure 2 right), it is not always attainable by deterministic classifiers.  Instead, we use a randomized classifier that interpolates between $g_1$ and $g_2$ to reach it.  Note that we can make $g_2$ deterministic by setting, e.g., $q=e_1$ (i.e., always output class 1).
>
> - **(Q5) "What's the issue to generalise it to attribute unaware case?"**
>
> 	If we have access to $A\mid X$, a way to tackle the attribute-unaware case may be to chain this with our procedure: given $X=x$, we first predict $A$ by taking $\hat a =\arg\max_a \Pr(A = a | X=x)$, then feed $(x,\hat a)$ into our classifier. This is indeed a possible extension, but will require additional analyses if we want to guarantee fairness and, especially, optimality; our analysis for the attribute-aware setting (for identifying the optimal fair classifier) relied on the fact that we can condition on the ground-truth sensitive attribute $A$. We leave this direction as future work.
>
> - **(Q6) "computation complexity of [the] method"**
>
> 	Since our implementation uses linear programs, the time complexity will depend on the specific solver used. To our knowledge, the best established rate for interior point methods is polynomial in the number of constraints and variables, giving us an overall $\widetilde O (poly(nk))$ time for our algorithm (lines 242-244).  E.g., if we apply the result in [1], we get $\widetilde O ((nk)^{2.5})$).
>
> - **(W1) "the [division] by p_{a, y} with large number of protected attribute and classes can make the bound non-informative"**
>
> 	To perform post-processing, we require data from all intersction of group and classes, $\mathcal A\times\mathcal Y$.  The term $p_{a, y}$ appears when the $n$ examples are drawn from the entire population without conditioning  (Assumption 4.1).  If we can sample conditionally and assume that every group-class combinations have at least $n$ examples, then the term can be dropped.  This term has also appeared in prior work, e.g., [2, 3].
>
> - **(W2) "the expectation in the fairness guarantee is not well placed"**
>
> 	Our (fairness) guarantee in Theorem 4.2 is a high probability bound, so it does not have an expectation operator.  And because it is a stronger result, it readily implies a bound on $\mathbb E[\Delta_\text{TPR}(h)]$.
>
> [1] Speeding-up linear programming using fast matrix multiplication. FOCS 1989.
> [2] Equality of Opportunity in Supervised Learning. NeurIPS 2016.
> [3] Learning Non-Discriminatory Predictors. COLT 2017.

---

> > ### Comment · Area_Chair_8jVU · 2023-08-11
> > **Follow-up on response to reviewer EVhV**
> >
> > If possible, please clarify the following points of confusion that remained after reading your rebuttal.
> >
> > Q1) The reviewer raised reference [A] as an essential point of comparison with your work. As mentioned by the reviewer, [A] only considers binary classification. However, it is important to note that, in the benchmarks provided in Alghamdi et al. [2], this method seems to perform very well. It is also important to acknowledge that [A] may be the first to use unlabelled data for post-processing. If possible, can you (qualitatively) compare your approach to the one in [A] and explain how you would address this in the paper?
> >
> > Q2) The runtime aspect is not transparently addressed in the manuscript, and I found the answer to Q6 insufficient. Though the proposed formulation solves a linear optimization -- for which complexity bounds are well understood, as noted in the rebuttal -- the reported runtimes in the SM are high. Specifically, the proposed method runs up to 50x slower than your reported benchmarks of FairProjection (Table 1, ~1.5min vs ~1h20min). I understand that FairProjection runs on GPU -- so this is not an equal comparison per se -- but while FairProjection is immediately parallelizable, it is unclear how the proposed approach could be sped-up beyond using a better LP solver. The trend in Table 1 is also worrying, with a significant increase in the runtime of the proposed method between ACSIncome and the BiasBios dataset.
> >
> > Can you comment more on the runtime issue and how you would address this in the numerical experiments/limitation section? I believe it is critical to be upfront about it.
> >
> > Q3) Finally, the paper mentions in lines 99-100 that [2] does not relate divergence to utility. However, the work [B] that predates [2] explicitly notes the connection between divergence measures and cross-entropy to log loss. In fact, [B] -- which FairProjection generalizes -- outperforms other binary fairness interventions when compared using score-based metrics (e.g., Brier Score). Could the fact that FairProjection optimizes for score-based variations of fairness metrics (vs. optimizing for thresholded scores) partly explain the discrepancy in performance in the "high fairness" regime in Figure 3? In other words, does FairProjection control for the same metric displayed in the figure? Or is FairProjection optimized for a different metric?
> >
> > If possible, please provide a more precise explanation of the difference between the optimization formulation in [2] with the one you introduce in Section 3, including differences in fairness metrics compatible with each approach (e.g., the difference between score-based/threshold-based formulations of fairness metrics, generalization beyond equalized odds, etc.).
> >
> > [B] Wei, D., Ramamurthy, K. N., & Calmon, F. P. (2021). Optimized score transformation for consistent fair classification. The Journal of Machine Learning Research, 22(1), 11692-11769.

---

> > > ### Author Response · Authors · 2023-08-13
> > > **Thank you area chair 8jVU**
> > >
> > > We thank the area chair for the review and comments, and hope that the following addresses the concerns:
> > >
> > > - **Q1.** We ran new experiments using the method proposed by [A] (referred to as `LevEqOpp`) on the ACSIncome dataset; the results are presented below. In comparison, our method can achieve slightly higher fairness, while LevEqOpp has slightly better tradeoff and lower variance (averaged over 10 seeds). The result will be added to our Fig. 3.
> > >
> > > 	We also thank the reviewer and the AC for the remark regarding [A], and will credit it on line 96 that “the first post-processing method for equal opportunity in the binary setting that also requires only unlabeled data is [A]”. We would like to clarify that we did not claim the need for just unlabeled data as a novelty (on lines 38–44) of our method, as we understand that there are many existing post-processing methods with the same data requirement.
> > >
> > > 	| ACSIncome (2 groups, 2 classes) | accuracy | $\Delta_\text{TPR}$ | runtime (seconds) |
> > > 	| --- | --- | --- | --- |
> > > 	| Ours (tolerance = 0.0001) | 0.7727 ± 0.0005 | 0.0060 ± 0.0031 | 139 |
> > > 	| Ours (tolerance = 0.005) | 0.7745 ± 0.0003 | 0.0087 ± 0.0024 | 119 |
> > > 	| Ours (tolerance = 0.008) | 0.7748 ± 0.0003 | 0.0099 ± 0.0013 | 119 |
> > > 	| LevEqOpp | 0.7748 ± 0.0004 | 0.0088 ± 0.0016 | 58 |
> > >
> > > - **Q2.** While our LP formulation and the use of (exact) LP solvers means that the only sources of (excess) error (in both fairness and suboptimality) are generalization and miscalibration because optimization error is negligible, you are correct in that it may have limitations in terms of scalability. We will highlight this limitation of LP solvers in Section 6, as well as to lines 686–688—which will be promoted to Section 5. We would like to add that our implementation runs on a single CPU core (which, a typical server CPU has ≥ 32 in total), so we parallelized runs with different random seeds and fairness tolerances to get the results in Fig. 3.
> > >
> > > 	We share some thoughts on the ways of overcoming this. Since Theorem 1.1 is a representation result for the optimal fair classifier, an alternative problem formulation is to directly search over function parameters $\beta$ and $\lambda$ for satisfying TPR parity (as a regularizer) while maximizing utility, and this could be done using mini-batch SGD in a parallelizable manner (details on, e.g., how to differentiate through argmax, could be worked out). To guarantee fairness, we would require convergence results; it is left as future work.
> > >
> > > - **Q3.** We thank the AC for the reference to [B] and the remarks on [2]!
> > >
> > > 	In terms of the discrepancy in performance, we will provide the explanation that [2] optimizes for log loss in Section 5 paragraph Results (also citing [B]), and revise the statements on lines 102–104. In terms of fairness, we will acknowledge that [2] aims for fairness of the scores. Regarding generalization to other fairness criteria, we will highlight on lines 99–100 that [2] is a general-purpose post-processing method applicable to criteria besides equal opportunity.
> > >
> > > 	While [2] is optimizing for log loss subject to fairness of the scores, i.e., $R\perp A\mid Y$ (although the evaluations performed in [2] are in terms of accuracy as the metric, and fairness of the class assignments), our method in Section 3 optimizes for classification accuracy subject to fairness of the class assignments ($\widehat Y \perp A\mid Y$). So in this regard, [2] and our paper are complementary in the fairness literature overall.
> > >
> > > 	Lastly, a unique benefit of our analysis and formulation is that, by focusing on TPR parity and establishing the representation result (Theorem 1.1), we are able to derive explicit guarantees for both the fairness and the optimality (Theorem 4.2) of the post-processed classifier produced by our method in Section 3.
> > >
> > > [2] Beyond Adult and COMPAS: Fair Multi-Class Prediction via Information Projection. NeurIPS 2022.

---

> > > > ### Comment · Area_Chair_8jVU · 2023-08-17
> > > > **Thank you for the clarification**
> > > >
> > > > Thank you for these clarifications -- they are helpful.
> > > >
> > > > I have one more clarification question:
> > > >
> > > > In your results, are the fairness criteria used in the benchmarked fairness interventions in Figure 3 (FairProjection and Adversarial) also TPR difference (Defn 2)? The authors of [2] present results in terms of what they call MEO, i.e., the worst-case average between TPR and FPR (see [2, Eq. 10]). In this regard, they seem to be controlling for a different type of fairness criteria that is more stringent than TPR difference. Similarly, the Adversarial approach seems to admit a range of fairness criteria which may not match exactly Defn 2.
> > > >
> > > > Are the FairProjection and Adversarial benchmarks implemented to control **exclusively** for the same TPR criteria as your approach, or are the benchmarked interventions controlling for a different fairness metric (e.g., a form of MEO in [Eq. 10, 2]), and being evaluated in terms of TPR difference? I want to understand if this is an apples-to-apples comparison.

---

> > > > > ### Author Response · Authors · 2023-08-18
> > > > > **Thank you area chair 8jVU**
> > > > >
> > > > > We thank the area chair for these questions, which we also put thoughts into during our literature review!
> > > > >
> > > > > The main points are stated first, followed by the supporting material:
> > > > >
> > > > > 1. From the source code provided by [2], we checked that the `meo` constraint in their code is equivalent to our TPR parity ($\Delta_\text{TPR}$, Definition 2). This means that `meo` (code) = TPR parity, and the experiments in [2] and our paper are indeed exclusively controlling for the same fairness metric.
> > > > >
> > > > > 2. We are aware that the $\textsf{MEO}$ metric defined in [2, Eq. (10)] is different from our TPR parity. But, from our reading—please kindly correct us if we misunderstood—we see that TPR parity = `meo` (in code) ≠ $\textsf{MEO}$ (in paper).
> > > > >
> > > > > 	- TPR parity (= `meo` in code, as mentioned above) is a relaxation of equalized odds (EqOdds, $\widehat Y\perp A\mid Y$). However, the $\textsf{MEO}$ metric defined in [2, Eq. (10)]—although motivated as such—is not a relaxation of EqOdds in **multiclass settings** (counterexamples provided below):
> > > > >
> > > > > 		$$\Delta_\text{TPR}=0\impliedby\Delta_\text{EqOdds}=0\centernot\implies\textsf{MEO}=0.$$
> > > > >
> > > > > 		This means that the discrepancy is, in fact, between the experiment setup of [2] and their evaluations.
> > > > >
> > > > > 	- The reason why this discrepancy did not seem to "negatively affect" the plots in [2] is because the term $|\text{TPR}_i(a)-\text{TPR}_i(a')|$ is typically larger than—hence dominates—$|\text{FPR}_i(a)-\text{FPR}_i(a')|$. Also, in binary classification, all three notions above are equivalent.
> > > > >
> > > > > 3. Adversarial debiasing is an in-processing method included to "serve as an upper bound", rather than to compete with (our) post-processing methods.
> > > > >
> > > > > 	It aims to learn representations $Z$ s.t. $Z\perp A\mid Y$, which would imply EqOdds and thereby TPR parity. While the EqOdds objective of adversarial debiasing is stronger than the TPR parity criterion of ours, to achieve EqOdds, one must first satisfy TPR parity, because $$\Delta_\text{TPR}\leq\Delta_\text{EqOdds}:=\max_{a,a'\in\mathcal A}\max_{y,y'\in\mathcal Y}|\mathbb P(\widehat Y=y,A=a\mid Y=y')-\mathbb P(\widehat Y=y,A=a'\mid Y=y')|.$$
> > > > >
> > > > > 	And, in practice, when a classifier has high accuracy but is unfair, $\Delta_\text{EqOdds}$ is typically attributed to $\Delta_\text{TPR}$.
> > > > >
> > > > > [2] Beyond Adult and COMPAS: Fair Multi-Class Prediction via Information Projection. NeurIPS 2022.
> > > > >
> > > > > ---
> > > > >
> > > > > ## Supporting material
> > > > >
> > > > > 1. In Table 2 and Appendix A.1.8 of [2], *(i)* the constraint and formula for EqOdds are provided; *(ii)* we get TPR parity by restricting that constraint to over $y=y'$ (notationally corresponds to their $c=c'$) instead of all pairs of $y,y'\in[k]$.
> > > > >
> > > > > 	Digging into their [source code](https://github.com/HsiangHsu/Fair-Projection/blob/1a1235f11f7b4564b0914b7e559892cbcc9f9d1e/fair-projection/enem/multi-group-multi-class/GroupFair.py#L150) for `meo`, we see that *(i)* is implemented on lines 172–178, and *(ii)* is reflected on line 150.
> > > > >
> > > > > 2. Recall from [2] that $$\begin{aligned}\text{TPR} _{i}(a)&=\mathbb P(\widehat Y=i\mid Y=i,A=a)=\frac{\mathbb P(\widehat Y=i,Y=i\mid A=a)}{\mathbb P(Y=i \mid A=a)},\\\\\text{FPR} _{i}(a)&=\mathbb P(\widehat Y=i\mid Y\neq i,A=a)=\frac{\mathbb P(\widehat Y=i,Y\neq i\mid A=a)}{\mathbb P(Y\neq i \mid A=a)}.\end{aligned}$$
> > > > >
> > > > > 	- Let $C_{a}\in\mathbb R^{k\times k}$ be the conditional confusion matrix on group $a$, i.e., $(C_a)_ {ji}:=\mathbb P(\widehat Y=i\mid Y=j,A=a)$, and $p_a$ the base rates, $(p_a)_j:=\mathbb P(Y=j\mid A=a)$. Then, $$\text{FPR}_i (a)=\frac{\sum _{j\neq i}(C_a) _{ji}(p_a)_j}{\sum _{j\neq i}(p_a)_j}.$$
> > > > >
> > > > > 		Consider a 3-class example where $\Delta_\text{TPR}=\Delta_\text{EqOdds}=0$: $$C_1=C_2=\begin{bmatrix}0.7&0.1&0.2\\\\0.1&0.6&0.3\\\\0.4&0.2&0.4\end{bmatrix},\text{ with }
> > > > > 		p_1=\begin{bmatrix}0.2\\\\0.7\\\\0.1\end{bmatrix},\ p_2=\begin{bmatrix}0.1\\\\0.1\\\\0.8\end{bmatrix},$$
> > > > >
> > > > > 		and note that none of the FPRs match:
> > > > >
> > > > > 		|Group|$\text{TPR}_1$|$\text{TPR}_2$|$\text{TPR}_3$|$\text{FPR}_1$|$\text{FPR}_2$|$\text{FPR}_3$|
> > > > > 		|-|-|-|-|-|-|-|
> > > > > 		|$A=0$|0.7|0.6|0.4|0.1375|0.1333|0.2778|
> > > > > 		|$A=1$|0.7|0.6|0.4|0.3667|0.1889|0.25|
> > > > >
> > > > > 	- (also point 3) When the number of classes is large and the classifier has good performance, the denominator of FPR is large and the numerator is small, so FPR will be much smaller than TPR. The same can be said of the off-diagonal terms in $C_a$'s considered by EqOdds.
> > > > >
> > > > > 		We present some numerical evidence on ACSIncome, where we practically see that $\Delta_\text{TPR}> \Delta_\text{FPR}$ in all cases, and $\Delta_\text{TPR}=\Delta_\text{EqOdds}$ except in the high TPR fairness regime achieved only by our method:
> > > > >
> > > > > 		|ACSIncome (5 groups, 5 classes; seed=33)|accuracy|$\Delta_\text{EqOdds}$|$\Delta_\text{TPR}$|$\Delta_\text{FPR}$|
> > > > > 		|-|-|-|-|-|
> > > > > 		|Adversarial ($\lambda=2$; MLP)|0.4592|0.3465|0.3465|0.1363|
> > > > > 		|No mitigation (logistic regression)|0.4529|0.6176|0.6176|0.2330|
> > > > > 		|FairProjection-KL ($\alpha=0$)|0.4496|0.2242|0.2242|0.1028|
> > > > > 		|Ours ($\alpha=0$)|0.4137|0.0798|0.0574|0.0458|

---

> > > > > > ### Comment · Area_Chair_8jVU · 2023-08-21
> > > > > >
> > > > > > Thank you for these clarifications and all the effort in the rebuttal -- this is very helpful. We will take all of this information into account in the discussion phase.

---

> > > > > > > ### Author Response · Authors · 2023-08-21
> > > > > > > **Thank you area chair 8jVU**
> > > > > > >
> > > > > > > We thank the area chair for the support, the time and effort in evaluating our work, and the helpful comments and thoughtful questions for improving the manuscript!

---

### Official Review · Reviewer_Wt5y · 2023-07-09

**Soundness:** 3 good
**Presentation:** 3 good
**Contribution:** 3 good
**Rating:** 7
**Confidence:** 3

**Summary:**


This paper studies algorithmic fairness in multiclass classification setting. The fairness notion considered is parity of true positive rates (TPR) which is the multi class analog to equalized odds. The paper gives a post-processing algorithm which, given a score function, outputs a fair classifier. The paper then gives sample and time complexity guarantees and experimental evaluation on benchmark datasets.

This paper furthers work studied Alghmadi et al., which according to the authors, is the only other post-processing method available for multi-class TPR parity.

The paper gives a general post-processing algorithm that takes as input a score function and outputs a classifier that satisfies approx TPR party. The authors show that if you begin with the Bayes score function, their post-processing returns an optimal, fair classifier. They also give results showing that if the initial score function is not Bayes optimal but instead satisfies a decision calibration condition, then the post-processing is optimal among all classifiers that can be derived from the initial score function.

The post-processing method consists of two parts. The first part of the process estimates the feasible region of TPRs and then finds the utility-maximizing TPRs that satisfy fairness constraints (either exactly via search if we know the distribution, or via a linear program if we are estimating the TPRs from data). The second step involves finding the hypothesis that achieves these TPRs, which either corresponds to a tilting - essentially a threshold - of the score function without randomization or a mixture of two models that lie on the boundary.

Finally, the paper gives some experimental evaluation on three benchmark datasets that serves as a proof of concept of the post-processing and showed that it is competitive with other standard techniques (notably reductions).

**Strengths:**

Fairness via post-processing in multiclass classification settings is not a focus on most prior work on algorithmic fairness, which makes this an interesting and novel contribution.

Although restricted to TPR parity, the paper does a complete analysis of the topic under this fairness definition. The paper addresses both not knowing the underlying distribution and so possibly starting from a non-Bayes score function and also discuss estimating the parameters from finite samples.

**Weaknesses:**

The paper gets a little notationally and technically dense in Section 3. While the presentation is still fairly clear, I think additional higher level exposition to describe in particular Step 2 of the algorithm could be beneficial to readers - maybe including additional description of a tilting.


**Questions:**

What are the assumptions on group structure? Must they be disjoint? This might be useful to clarify.


**Limitations:**

The authors address one of the main limitations of this work that their method is confined to only reducing TPR disparity and not other fairness notions.

---

> ### Author Rebuttal · Authors · 2023-08-10
>
> We thank the reviewer for the time, comments, and support! We hope that the following addresses the concerns:
>
> - **"What are the assumptions on group structure? Must they be disjoint?"**
>
> 	We do not impose any requirement on the group structure; the practitioner shall set $\mathcal A$ to include all (combinations of) groups with respect to which equal opportunity is demanded.
>
> 	For instance, if there are two sensitive attributes $\mathcal A_1$ and $\mathcal A_2$, and we require fairness w.r.t. (sub)groups defined at all levels (i.e., intersecting/overlapping), then we set $\mathcal A=\mathcal A_1\times \mathcal A_2$.
>
> - **"The paper gets a little notationally and technically dense in Section 3..."**
>
> 	Thank you for the suggestions!  We will include more discussions, incl. high-level descriptions of the algorithm, in section 3.

---

### Decision · Program_Chairs · 2023-09-21

**Decision:**

Reject

**Comment:**

The paper describes an interesting post-processing method for closing TPR gaps. The post-processing method performs well on numerical benchmarks when using well-calibrated base classifiers. The paper does miss important related literature --- an issue raised by most reviewers --- but I don't hold that directly against it. That's what reviews are for.

The paper was subject to debate, both in the AC-reviewer discussion and in discussion with the SAC. The AC initially had a more favorable view of the paper but was persuaded by limitations noted by the reviewers during the discussion. Ultimately, the rejection decision was due to the paper having several (potentially fixable) limitations that, together, amount to a significant revision of the manuscript and would require a new review.

The main issues were:

**Limitations in the presentation of theoretical results:** There were issues with the presentation of technical results. Notably, one reviewer found that the statement of Theorem 3.5--the main theoretical result---has two versions: one in the main paper and one in the SM, and they don't match. Please see below a quote the reviewer sent the AC and SAC. There are also issues with notation and presentation being inconsistent in the main text and in the SM. One reviewer also noted that a theorem they requested in their review "appeared" in the rebuttal, but to ensure correctness would require a new round of review. Overall, these bugs can potentially be fixed but would require a new review since they impact the main text.

***Limitations in "efficiency" claims:** One reviewer noted that the first word in the title is "efficient," but the authors only present runtime benchmarks in the SM. Upon inspection, their method runs 50x slower than competing benchmarks (taking hours instead of minutes).The authors rebuttal to this point was that the method requires running an LP, so it runs in polynomial time. During the discussion, one reviewer did not find that an issue: "LP is as efficient as it gets." However, other reviewers disagreed. Not all LPs are created equal, and one where the number of optimization variables scales with the number of samples in a dataset is particularly inefficient. The authors suggested methods to overcome this (e.g., early stopping of a solver) which could work, but would require a new round of review.

**Issues with assumptions:** The method inherently depends on a well-calibrated base model and may not work if this is not the case. Some reviewers found this a significant limitation. The authors primarily employ a linear logistic regression model in their experiments due to its inherent calibration benefits. However, this choice limits their method's accuracy, preventing it from outperforming the Adversarial technique that uses MLPs. Reviewers found this calibration requirement presents significant theoretical and practical challenges for their approach. The authors briefly mention this limitation, but it needs more discussion, which, in turn, would require a new round of review.

**Issues with benchmark methods:** Reviewers raised concerns surrounding benchmarks against competing methods. The Adversarial method controls for EqOpp and can be used with MLPs, achieving higher accuracy. FairProjection has greater flexibility, seemingly being applicable to any probabilistic classifier and controlling for more fairness metrics. Nevertheless, they are benchmarked in the unique setting the proposed intervention is optimized for. Reviewers noted that the paper needs to be more upfront about this. This update would also require a new round of review.

In my view, this is a clear "major revision" if this were a journal since the updates, though feasible, are significant and would change the main body of the manuscript and require a new round of review. Unfortunately, since this is not possible at NeurIPS, the decision (after discussion with the SAC) is reject.

---
**Comments sent to AC/SAC from a reviewer:**

The main theoretical result is Theorem 3.5, which has two versions: one in the originally submitted manuscript and one in the version with the SM which is significantly changed. In the latter version, there is an additional assumption of "finer granularity" and the upper bounds are different. Even with the newer version, a close inspection of the proof reveals that the authors overlooked the necessity of the assumption of nonnegativity of the vector "v" (see, e.g., the steps at the top of page 22 in the SM), which makes claims in this proof such as "the other side follows symmetrically" questionable (Line 630). Other examples include: the objective in LP3 (Lines 239-240) doesn't make sense; the probability statement is missing in Theorem 4.2; the term "beta" is used in almost all lemmas in the appendix as a number in the statements and then in the proof as a vector with some properties. It is ok for a paper submission to have several typos, but for the errors to appear in the result statements makes assessing the correctness impossible.